# Is Heterogeneity Notorious? Taming Heterogeneity to Handle Test-Time Shift in Federated Learning

**Yue Tan**[*]
University of Technology Sydney
yue.tan@student.uts.edu.au

**Chen Chen**
Sony AI
chena.chen@sony.com

**Weiming Zhuang**
Sony AI
weiming.zhuang@sony.com

**Xin Dong**
Sony AI
xindong@sony.com

**Lingjuan Lyu**[†]
Sony AI
lingjuan.lv@sony.com

**Guodong Long**
University of Technology Sydney
guodong.long@uts.edu.au

## Abstract

Federated learning (FL) is an effective machine learning paradigm where multiple clients can train models based on heterogeneous data in a decentralized manner without accessing their private data. However, existing FL systems undergo performance deterioration due to feature-level test-time shifts, which are well investigated in centralized settings but rarely studied in FL. The common non-IID issue in FL usually refers to inter-client heterogeneity during training phase, while the test-time shift refers to the intra-client heterogeneity during test phase. Although the former is always deemed to be notorious for FL, there is still a wealth of useful information delivered by heterogeneous data sources, which may potentially help alleviate the latter issue. To explore the possibility of using inter-client heterogeneity in handling intra-client heterogeneity, we firstly propose a contrastive learning-based FL framework, namely FedICON, to capture invariant knowledge among heterogeneous clients and consistently tune the model to adapt to test data. In FedICON, each client performs sample-wise supervised contrastive learning during the local training phase, which enhances sample-wise invariance encoding ability. Through global aggregation, the invariance extraction ability can be mutually boosted among inter-client heterogeneity. During the test phase, our test-time adaptation procedure leverages unsupervised contrastive learning to guide the model to smoothly generalize to test data under intra-client heterogeneity. Extensive experiments validate the effectiveness of the proposed FedICON in taming heterogeneity to handle test-time shift problems.

## 1 Introduction

Federated learning (FL) is a promising machine learning paradigm that enables multiple clients or devices to collaboratively train models without sacrificing their data privacy [1, 2]. Practical FL frameworks usually suffer from notorious non-IID (independent and identically distributed) local data distributions, which may lead to severe performance degradation during the test phase [3, 4].

---

[*]Work done during Yue Tan's internship at Sony AI.
[†]Corresponding Author.

37th Conference on Neural Information Processing Systems (NeurIPS 2023).

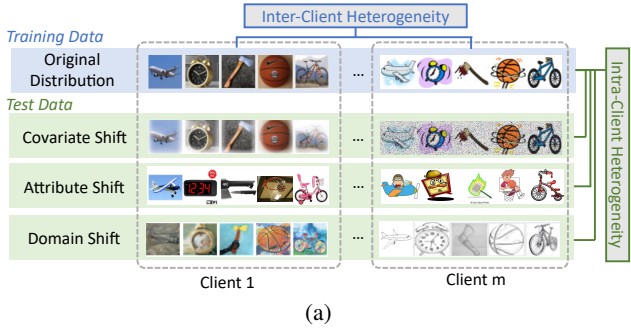 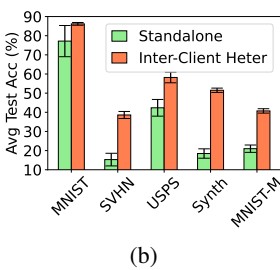

(a)

(b)

Figure 1: (a) Illustration of our setting where both *inter-client heterogeneity* and *intra-client heterogeneity* exist in an FL framework. (b) Compared with training on data from a single domain, the improved robustness to test-time shift is achieved when training on heterogeneous data sources, indicating that the inherent *inter-client heterogeneity* in FL has a positive effect in handling test-time shift problems. Detailed settings can be found in Appendix A.1.

To improve the performance of the trained model on local data distribution, there are a variety of personalized federated learning (PFL) methods based on different techniques, *i.e.*, regularization [5, 6], meta-learning [7, 8], knowledge distillation [9, 10], to help the model adapt to local datasets [11].

In existing PFL systems, it is commonly assumed that the training and test datasets within a client share the same or similar distribution. However, this assumption often does not hold in real-world applications where test samples may encounter natural variations or corruptions. Although test-time distribution shifts have received increasing attention in centralized machine learning paradigms [12, 13, 14, 15], the negative effects of these shifts in PFL systems are largely underexplored. Take federated chest X-ray classification [16, 17] among multiple hospitals as an example, while most existing studies acknowledge that different hospitals can have varying data distributions due to diverse patient populations (e.g., races, ages, etc.), they overlook the possibility of variation within a single hospital. For instance, there may be multiple X-ray machines from different vendors, each with its own unique internal parameters (e.g., exposure time, spectrum, etc.). Similar issues can be found in various other domains, such as autonomous vehicles and video surveillance. Consequently, an efficient and effective solution to handle test-time shifts is necessary to achieve satisfactory performance.

The extensively studied heterogeneous distribution in FL, also known as non-IID issue, mainly refers to the *inter-client heterogeneity* where clients own heterogeneous training data during their federated training phase [2, 3, 18]. Differently, the aforementioned test-time distribution shift during the inference stage of PFL refers to the heterogeneity within a client, which can be defined as *intra-client heterogeneity*. As illustrated in Fig. 1(a), the test data of a client can differ from its training data in various aspects. This discrepancy results in unexpected features in the model optimized on the training data, ultimately leading to suboptimal performance. In this case, a natural question that arises is: *what is the difference between the test-time shift in centralized settings and that in FL?* Interestingly, we find that the answer lies in the unique inherent inter-client heterogeneity in FL compared with centralized settings. While *inter-client heterogeneity* is often considered to be a challenge for FL, in this paper, we pinpoint that heterogeneous sources can also convey a wealth of useful information that potentially helps alleviate test-time shift problems in FL systems. A benefit of this heterogeneity is that it enables the global model to capture robust features that are less sensitive to both inter-client and intra-client shifts. We firstly discover that this information has the potential to help alleviate test-time shift problems in FL systems, as it enables the global model to capture robust features that are less sensitive to both inter-client and intra-client shifts. Through our motivated experiments (Fig. 1(b)), we observe that FL under *inter-client heterogeneity* can improve the ability of clients to handle test-time shift problems compared to training on homogeneous data. In this way, we turn the problem of inter-client heterogeneity as a blessing in solving test-time shift.

Motivated by the above findings, we introduce **Fed**erated Learning via **I**nvarian**C**e Extracti**O**n and Shari**N**g (FedICON), a contrastive learning-based FL approach that enables clients to learn the robust features on heterogeneous data. Our theme is to fully take advantage of the benefit brought by *inter-client heterogeneity* during training time, and then, at the test phase, tune the local model of each client on its private test data to conquer the *intra-client heterogeneity*. During the training phase, clients first carry out local representation learning under the guidance of label information

so that invariant information underlying each training sample, as well as the class it belongs to, is extracted. At the global aggregation stage of federated training, the parameters of feature encoders in all clients are aggregated to further boost the ability of clients to learn invariant information among inter-client heterogeneity. Then, during the test phase, we introduce self-supervision signals to the unlabeled test samples and smoothly tune the feature encoder based on the acquired knowledge in identifying invariant properties between diverse training data and shifted test data. Notably, we leverage contrastive learning paradigms for both federated training and local test-time adaptation, enabling the trained model to well adapt to test data distribution.

Our contributions are summarized as follows:

- We are the first to comprehensively investigate feature-level test-time shifts in FL. We formulate the problem from the perspective of *inter-client heterogeneity* and *intra-client heterogeneity* and propose corresponding experimental settings tailored for this problem.

- We propose a novel representation learning-based federated learning algorithm, termed FedICON, that addresses the test-time shift problem by taming inherent heterogeneity in FL for invariance extraction and sharing.

- We conduct extensive experiments on benchmark datasets to measure the performance of FedICON in handling feature-level test-time shift problems. The results demonstrate that FedICON achieves significant improvement in terms of test-time adaptation ability.

## 2   Related Work

**Personalized Federated Learning.**   Personalized federated learning (PFL) aims to enable each client to learn a personalized model that performs well on its local dataset [19, 11]. Existing PFL methods are developed on the basis of various techniques. A branch of work utilizes model decoupling to separate the local model into the globally shared part and personally trained part to realize different levels of personalization [20, 21, 22, 23, 24]. [25, 26, 27, 5, 28] augment the original objective function with additional regularization terms to induce personalization via representation alignment, knowledge transfer, bi-level optimization, etc. [8, 7] incorporate meta-learning to generate the generalized model which can be further optimized as a personalized model. Existing personalized federated learning methods can only personalize the model to local training data, lacking the ability to generalize to the test dataset [29].

**Federated Learning with Feature-Level Distribution Shifts.**   Non-IID issue is the core challenge in both FL and PFL [30]. A common type of non-IID data in real-world scenarios is known as feature-level distribution shifts, including attribute shift, covariate shift, and domain shift [31, 32]. For instance, in FL, the training and test images can be collected in different time period, differnt locations, and even from different photographic equipment. [33] leverages mutual information theory and diversity augmentation to mitigate the negative effects brought by attribute shift and domain shift. [18] proposes to use local batch normalization to alleviate the feature-level distribution shift problems. Enlightened by the idea of domain generalization methods, [34, 35] design different knowledge transfer strategies to learn high-quality domain consensus knowledge across shifted clients. Based on feature-level distribution shift setting, [36, 37] aim to alleviate the inter-domain knowledge forgetting problem and performance degradation problem on unseen clients/domains, respectively.

**Test-Time Adaptation.**   Test-time adaptation methods aim to improve the model performance on out-of-distribution data by adapting the learned model to test data [38, 39]. Several methods propose to adjust the model by minimizing the pre-defined test-time entropy in various forms [40, 15]. [41] presents a classifier-only adjustment module for test domain adaptation, and [12] further shows that selectively fine-tuning part of the model can address different types of distribution shifts. [13, 42] tune the model on unlabelled test data via self-supervised learning objectives. [43] further proposes to use more informative self-supervised learning to solve the test-time performance degradation in the presence of severe distribution shifts. The performance of existing centralized test-time adaptation methods can be hindered due to the inherent heterogeneity in FL during the training phase, which further affects the test adaptation process.

**Federated Representation Learning.**   Representation learning focuses on extracting higher-level representations from raw data using deep neural networks [44]. Combining it with the federated

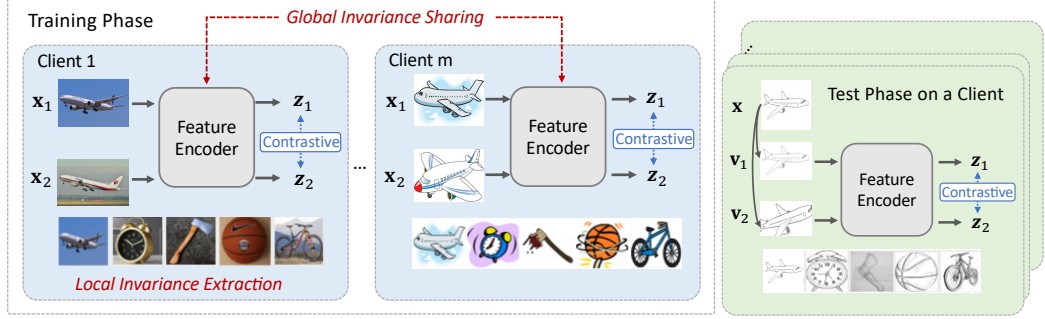

(a) Federated Training under Inter-Client Heterogeneity    (b) Test-Time Adaptation

Figure 2: An overview of the proposed FedICON. (a) Federated training phase under inter-client heterogeneity. Each client conducts representation-wise local invariance extraction during local training and performs parameter-wise global invariance sharing during global aggregation. (b) Test phase on a single client. Clients carry out independent test-time adaptation to generalize to their local test sets.

learning framework enables distributed clients to learn powerful representation and thus benefit their downstream tasks [45, 46, 47, 48]. [49] introduces a generalized federated self-supervised learning framework and provides in-depth empirical studies on the effect of key components in federated representation learning. [50] incorporates model decoupling and representation learning for better PFL. [51, 52] aim to learn a common representation model without supervision under the FL framework. Both of them utilize Siamese networks as their local representation module.

## 3 Notations and Preliminaries

### 3.1 Notations

Considering $m$ clients in an FL system, $P_i$ and $Q_i$ are training and test data distributions of the $i$-th client ($i \in \{1, \cdots, m\}$) respectively and $\mathcal{D}_i^P$ and $\mathcal{D}_i^Q$ are the corresponding training dataset and test dataset. A sample-label pair is denoted as $(\mathbf{x}, y)$ where $\mathbf{x}$ is the raw input data sample and $y$ is its corresponding label.

### 3.2 Inter-Client Heterogeneity

For arbitrary client $i$ and client $j$ in an FL system, the *inter-client heterogeneity* refers to varying $P_i$ and $P_j$ with respect to their raw data sample $\mathbf{x}$ and/or label $y$. For example, in the cases where $P_i(y) \neq P_j(y)$ while $P_i(\mathbf{x}|y) = P_j(\mathbf{x}|y)$, there exists an inter-client label distribution shift. The case where $P_i(\mathbf{x}) \neq P_j(\mathbf{x})$ while $P_i(y|\mathbf{x}) = P_j(y|\mathbf{x})$ can lead to inter-client feature distribution shift, which is common in real-world FL systems [18, 33, 36].

### 3.3 Intra-Client Heterogeneity

Most existing works on FL consider the heterogeneity of training datasets $\mathcal{D}_i^P$ across clients while overlooking the intra-client heterogeneity $P_i \neq Q_i$. [31] examines test distribution in robust FL benchmarks, combining label-level and feature-level shifts. To better understand their distinct impacts, we propose to study these shifts separately. Here, we focus on varying $P_i$ and $Q_i$ with respect to $\mathbf{x}$ rather than $y$, i.e., addressing the feature-level test-time shift problem in FL. Incorporating real-world scenarios, we thoroughly examine inter-client feature distribution shifts, including covariate shift, attribute shift, and domain shift. We assume that all clients have access to their own *labelled* training data when participating in the federated training process and test the performance of their personally tuned model on their own test set.

**Challenges.** It is a challenging task to deal with both inter-client and intra-client heterogeneity in an FL system. Although there is useful information underlying heterogeneous clients, how to effectively extract it and how to further leverage it to benefit the test procedure remain unexplored. Hence, there is a need for feasible solutions to tame inherent inter-heterogeneity to handle the test-time intra-client heterogeneity in FL.

# 4 Federated Learning via Invariance Extraction and Sharing

Both inter-client heterogeneity and intra-client heterogeneity arise from shifts in data distribution. Our core idea is to take advantage of diverse data distributions across clients so that robust features that remain invariant can be learned to handle intra-client shifts. Motivated by this, we propose **Fed**erated Learning via **I**nvarian**C**e Extracti**O**n and Shari**N**g (FedICON) as shown in Figure 2. During the federated training phase, to extract invariant information of each class within the local dataset, we design a *local invariance extraction* scheme by employing contrastive learning in a supervised manner (Sec. 4.1.1). Then, we perform *global invariance sharing* under inter-client heterogeneity to further boost the invariance extraction ability for each client (Sec. 4.1.2). During the test phase, each client tunes the personalized model to adapt to its local test set (Sec. 4.2).

**Optimization Objective.** Since each client has varying training and test data distribution, the global objective of the personalized FL system across $m$ clients can be formulated as a two-stage optimization process,

$$(\theta_1', \theta_2', \cdots, \theta_m') = \min_{(\theta_1, \theta_2, \cdots, \theta_m)} \frac{1}{m} \sum_{i=1}^{m} \frac{|\mathcal{D}_i^P|}{N^P} L_{i,\text{sup}}\left(\theta_i; \mathcal{D}_i^P\right), \qquad (1)$$

$$\theta_i^* = \min_{\theta_i} L_{i,\text{us}}\left(\theta_i; \mathcal{D}_i^Q\right), \qquad (2)$$

where Eq. 1 and Eq. 2 are the optimization objective during the federated training phase and test phase, respectively. In Eq. 1, $L_{i,\text{sup}}$ and $\theta_i$ are the local supervised loss function and learnable model parameters for client $i$, respectively. $N^P$ is the total number of training samples among all clients. In Eq. 2, $\theta_i$ is initialized as $\theta_i'$ and further optimized on the test data of client $i$ via unsupervised learning loss function $L_{i,\text{us}}$. The eventual purpose of the framework is to efficiently learn invariant knowledge under inter-client heterogeneity during the training phase so that the intra-client heterogeneity can be alleviated via further optimization during the test phase.

## 4.1 Federated Training under Inter-Client Heterogeneity

Compared with the test-time shift problem in centralized settings, there are inherent heterogeneous source data in the federated learning framework, which is of great potential to utilize to deal with the shifted test data. Although the inter-client heterogeneous data are not directly accessible due to privacy concerns [53], their underlying invariant information can be captured by each client during local invariance extraction procedure and then globally shared across all clients to further enhance the invariance extraction ability.

### 4.1.1 Local Invariance Extraction.

To capture invariant information from samples within the same class, we design a local invariance extraction procedure based on supervised contrastive learning. As shown in Fig. 2(a), samples are first fed into a feature encoder to generate the representation for each sample. Then, a contrastive learning scheme is applied to the representations under the guidance of their corresponding labels so that the class-level invariant information can be extracted during the local training phase.

To make the representation ability of the feature encoder more robust, we add an augmentation module during the data pre-processing stage that incorporates some perturbations to raw samples. In this way, there are various views of the same sample fed into the feature encoder to strengthen its sample-based discrimination ability. Concretely, for the $i$-th client, given a batch of input samples, we generate two random augmented views for each sample. The total size of the augmented local dataset is $2|\mathcal{D}_i|$. For a specific sample $\mathbf{x}$, we use $A(\mathbf{x})$ to denote the local dataset without $\mathbf{x}$.

Taking advantage of the powerful representation ability brought by contrastive learning frameworks, we force the feature encoder to discriminate the augmented samples from different classes. Given an input sample $\mathbf{x}$, the output of the feature encoder is

$$z(\mathbf{x}) = h\left(\mathbf{x}; \theta_i\right). \qquad (3)$$

where $\theta_i$ is the learnable parameter of the feature encoder. We denote $z(\mathbf{x})$ as $z_{\mathbf{x}}$ for short. Samples from the same class are forced to be closer, and samples from different classes are forced to be farther.

We perform local training with the loss function as below,

$$L_{i,\text{sup}}\left(\theta_i; \mathcal{D}_i^P\right) = \sum_{(\mathbf{x},y)\in\mathcal{D}_i^P} \frac{-1}{|P(\mathbf{x})|} \sum_{\mathbf{p}\in P(\mathbf{x})} \log \frac{\exp\left(z_\mathbf{x} \cdot z_\mathbf{p}/\tau\right)}{\sum_{\mathbf{a}\in A(\mathbf{x})} \exp\left(z_\mathbf{x} \cdot z_\mathbf{a}/\tau\right)}, \tag{4}$$

where $z_\mathbf{a}$ is the representation of sample $\mathbf{a} \in A(\mathbf{x})$, and $\tau$ is the temperature adjusting the tolerance for representation difference. $P(\mathbf{x})$ is the set of samples that have the same labels with $\mathbf{x}$ and can be represented as

$$P(\mathbf{x}) = \{\mathbf{p} \in A(\mathbf{x}) : y_\mathbf{p} = y_\mathbf{x}\}. \tag{5}$$

By training the local feature encoder in a contrastive manner, the local invariances can be directly learned from the augmented data and be encoded to the feature encoder. Even if there is inter-client heterogeneity among different clients, their invariance encoding abilities over heterogeneous data are universal, which makes it possible for clients to mutually boost their performance on invariance extraction in a federated framework.

### 4.1.2 Global Invariance Sharing.

To tame the inter-client heterogeneity during the federated training process, there should be a uniform information channel that is globally shared across clients and continuously provides meaningful knowledge to boost local invariance extraction ability. Considering the fact that the feature encoder introduced above is trained to capture and encode the invariance, we can employ the parameters in the feature encoder as the information carrier and perform global invariance sharing at the server side. These invariant robust properties captured by the feature encoder are shared within different augmented views of a specific sample and different samples from the same class.

Similar to the standard federated aggregation process, for each client $i \in \{1, 2, \cdots, m\}$, the learnable parameter $\theta_i$ of its feature encoder is sent to the server. After receiving parameters from all the clients, the server aggregates them as

$$\bar{\theta} = \frac{1}{m} \sum_{i=1}^{m} \frac{|\mathcal{D}_i^P|}{N^P} \theta_i, \tag{6}$$

and sends $\bar{\theta}$ back to each client as their initialized parameter in a new round. Since the feature encoder of each client is trained to extract invariance efficiently rather than classify the samples correctly, the parameter aggregation is less likely to be affected by inter-client heterogeneity. Global invariance sharing with respect to the feature encoder incorporates the invariance encoding abilities learned upon heterogeneous source data, which reversely provides sufficient homogeneous information for each client and increases the robustness of local representation learning. To employ the model trained alternately via local invariance extraction and global invariance sharing, we further train a linear classifier on top of the fixed feature encoder [54, 55]. The linear classifier can be trained in a pure local manner or trained in a federated learning manner. In this paper, we conduct multiple federated training steps at first and then personalize the classifier for each client.

## 4.2 Test-Time Adaptation to Handle Intra-Client Heterogeneity

In Section 4.1, we have developed components that leverage the inherent inter-client heterogeneity in FL in order to handle the test-time intra-client heterogeneity. To keep the losses of the training and test phase unified and make the whole process consistent, we adopt a self-supervised contrastive learning scheme to carry out test-time adaptation, which is also a natural and efficient way to generalize the trained model to the unlabeled and shifted test set.

Concretely, for client $i$, the trained feature encoder is further tuned by optimizing the following unsupervised loss function on its local test set $\mathcal{D}_i^Q$.

$$L_{i,\text{us}}\left(\theta_i; \mathcal{D}_i^Q\right) = \sum_{\mathbf{x}\in\mathcal{D}_i^Q} -\log \frac{\exp\left(z_\mathbf{x} \cdot z_\mathbf{x}'/\tau\right)}{\sum_{a\in A(\mathbf{x})} \exp\left(z_\mathbf{x} \cdot z_\mathbf{a}/\tau\right)}. \tag{7}$$

where $z_\mathbf{x}'$ is the representation of the augmented sample of $\mathbf{x}$. Similar to the training phase, $A(\mathbf{x})$ here also denotes the local test dataset without $\mathbf{x}$. The main difference between Eq. 4 and 7 is whether the label information is accessible or not. Apart from this, Eq. 4 and 7 are consistently formulated and

can share the same optimization strategy. During the training phase, the contrastive learning process can be conducted under the guidance of label information, which encourages the feature encoder to output well-aligned representations for samples within the same class. Then, based on the robust representations learned among inter-client heterogeneity, the feature encoder can further improve the generalization ability on the test set in an unsupervised manner, which preserves the contrastive learning mode in the training phase.

The test scenario can be easily extended to the online manner where the test samples arrive one by one. First, a set of training samples is selected from the training set, replacing the $A(\mathbf{x})$ in Eq. 7. Then, after performing the optimization for each test sample, we replace one randomly selected training sample in $A(\mathbf{x})$ with this test sample for future test samples. This transfers the whole pipeline into an online manner and allows the model to gradually adapt to the distribution of test data.

We present our proposed FedICON in detail in Algorithm 1 in Appendix C.

# 5 Experiments

## 5.1 Experimental Setup

**Datasets and test-time shift settings.** We evaluate our proposed method on the following four benchmark datasets: Digit-5 [56], Office-10 [57], DomainNet dataset [58], and CIFAR10 [59]. Digit-5, Office-10, and DomainNet dataset are three benchmark dataset consisting of multiple sub-datasets from different domains. We use them to simulate the *covariate shift* and *domain shift* during the test phase. We use CIFAR10 and its variant CIFAR10.1 [60, 61] to simulate the *attribute shift*. Details about the test-time shift settings can be found in Appendix A.2.

- Covariate shift. We follow the setting in [31] by leveraging 15 common corruptions in CIFAR10-C [62], *e.g.*, Gaussian Noise, Brightness, etc.
- Attribute shift. The training data of all clients are from CIFAR10, while the test data of all clients are from CIFAR10.1 shifted from CIFAR10, *e.g.*, the most common object in the airplane class can change from airliner to fighter.
- Domain shift. We follow the training data setting in [18], where clients own data from different domains with heterogeneous appearances. We follow the leave-one-out principle to select one domain as the test domain and the other domains as the training domains. The training data of each client is from a single domain.

**Baselines and implementation.** We first compare our proposed method with two kinds of baselines: 1) Local, where each client only performs local training without any communication; FL/PFL methods including 2) FedAvg [63], 3) FedAvg-FT that fine-tunes the global model trained by FedAvg on local dataset, 4) FedProx [64], 5) FedRep [22], 6) FedBN [18], 7) FedTHE [31]. We also compare with three classical centralized test-time adaptation (TTA) methods, including 1) Tent [38], 2) EATA [40], and 3) T3A [41]. In practice, we implement these TTA methods with the FedAvg framework. Details about the baseline methods and their parameter settings can be found in Appendix A.3.

For Digit-5, Office-10, and CIFAR10, we use simple CNN architecture as the backbone model, while for DomainNet dataset, we use ResNet18 as the backbone model. Details about the model of each dataset can be found in Appendix A.4. We set the local training batch size as 32, and an SGD optimizer with weight decay rate 5e-4 and learning rate 0.01 for cross-entropy loss and 0.005 for contrastive loss. The temperature in Eq. 4 and Eq. 7 is 0.1 for Digit-5, Office-10, CIFAR10, 10 for DomainNet dataset. The size of representation used for contrastive learning is 64 for Digit-5, Office-10, and CIFAR10, and 512 for DomainNet dataset. The default number of local epochs is $E = 5$ and the number of communication rounds is 100. We implement all the methods using PyTorch and conduct all experiments on NVIDIA Tesla V100 GPU. Detailed settings can be found in Appendix A.

## 5.2 Performance Comparison

**Performance on various test-time shifts.** Table 1 and Table 2 report the test accuracy of FedICON and baselines on Digit-5 dataset under *covariate* test-time shift and Office-10 dataset under *domain* test-time shift, respectively. The results are reported in mean(standard error) format over three independent runs.

Table 1: Test accuracy on Digit-5 dataset under *covariate* test-time shift. All results are reported in mean(std error) format over three independent runs.

| Method | MNIST | SVHN | USPS | Synth | MNIST-M |
|--------|-------|------|------|-------|---------|
| Local | 60.65(4.79) | 34.43(1.87) | 16.91(4.55) | 39.93(3.25) | 18.07(3.26) |
| FedAvg | 86.18(0.72) | 38.60(1.84) | 58.19(2.78) | 51.50(1.13) | 40.72(1.15) |
| FedAvg-FT | 85.92(1.14) | 40.25(1.47) | 54.28(1.90) | 51.23(0.72) | 41.08(1.21) |
| FedProx | 86.45(0.83) | 38.95(2.29) | 57.80(2.52) | 52.40(1.32) | 40.55(0.48) |
| FedRep | 66.78(6.40) | 44.17(3.29) | 15.81(5.84) | 44.60(2.40) | 19.33(3.08) |
| FedBN | 61.42(4.50) | 41.38(0.77) | 41.73(7.44) | 42.77(4.44) | 42.35(1.13) |
| FedTHE | 84.52(0.21) | 39.00(2.21) | 58.03(2.96) | 51.92(1.27) | 40.55(1.23) |
| Tent | 86.13(0.56) | 43.87(0.81) | 67.51(2.40) | 50.47(0.12) | 47.73(0.94) |
| EATA | 85.37(0.08) | 43.65(1.55) | 67.12(1.48) | 50.70(0.22) | 47.58(0.77) |
| T3A | 72.58(2.50) | 44.23(1.77) | 49.44(5.63) | 45.62(0.95) | 46.13(1.53) |
| FedICON | **89.67**(1.48) | **45.23**(0.35) | **72.13**(2.22) | **56.13**(0.63) | **47.98**(0.50) |

The results suggest that our proposed FedICON outperforms existing FL/PFL methods and test-time adaptation methods and shows a strong ability to deal with both covariate and domain test-time shift problem. It should be noticed that although FedRep and FedBN are two state-of-the-art solutions in personalized federated learning, they usually fail to perform well on the shifted test sets. The main reason for that is both of them have a special focus on local training data distribution and accordingly adjust part of their model to better fit local training data distribution, leading to a more biased model than the generic model during test phase.

Table 2: Test accuracy on Office-10 dataset under *domain* test-time shift. All results are reported in mean(std error) format over three independent runs.

| Method | Test Domain | | | |
|--------|-------------|---------|---------|---------|
| | Amazon | Caltech | DSLR | Webcam |
| Local | 24.07(0.61) | 18.62(0.31) | 36.46(2.76) | 28.81(2.04) |
| FedAvg | 44.91(2.99) | 26.27(1.63) | 61.46(3.12) | 48.21(3.76) |
| FedAvg-FT | 41.78(1.23) | 24.94(1.07) | 55.56(2.41) | 46.52(5.72) |
| FedProx | 42.53(1.76) | 24.44(0.93) | 56.60(1.20) | 42.94(3.53) |
| FedRep | 20.02(1.16) | 17.68(1.34) | 34.03(5.92) | 24.86(1.49) |
| FedBN | 42.53(2.18) | 28.64(0.62) | 51.04(5.41) | 43.69(1.63) |
| FedTHE | 41.61(1.61) | 24.79(0.95) | 56.94(2.17) | 44.63(1.79) |
| Tent | 38.25(1.92) | 25.93(2.57) | 54.17(1.04) | 42.00(1.98) |
| EATA | 38.25(2.03) | 25.73(2.59) | 54.86(0.60) | 42.75(1.18) |
| T3A | 39.76(0.52) | 22.77(1.20) | 53.82(0.60) | 45.76(2.87) |
| FedICON | **50.46**(0.78) | **33.28**(0.67) | **67.01**(1.59) | **48.21**(2.67) |

Table 3 summarizes the results of test accuracy on CIFAR10 dataset under *attribute* test-time shift. We follow the setting of the naturally shifted test in [31]. The results in Table 3 indicate that our proposed FedICON is an effective solution to extract key invariant implicit information underlying both the training and test set, even though there are explicitly different properties conveyed by the training and test set. We also report how the performance varies on the original test set and the attribute test-time shift scenarios. The results show that FedICON has the smallest performance difference among all federated methods, which reflects the robustness of FedICON against attribute shift during test time. Additional experimental results on other datasets under various test-time shifts can be found in Appendix B.

**Effect of varying numbers of participating clients.** To test the performance of FedICON in a larger FL system, we increase the number of clients from 5 to 80 on Digit-5 dataset under *covariate* test-time shift setting. The total number of training samples is kept constant. As the number of clients becomes larger, the number of samples in each client becomes fewer. As shown in Fig. 3(a), both FedAvg and FedICON have a decreased performance when there are more clients as a result of the further divergence among participating clients. Although performance degradation is inevitable, FedICON still stably outperforms FedAvg in FL systems at different scales, demonstrating the scalability of FedICON and its ability to adapt to a large-scale FL system.

**Effect of varying numbers of local epochs.** To test how local epoch number affects the test performance, we vary the number of local epochs and report the results of FedAvg and FedICON on Digit-5 dataset under *covariate* test-time shift setting. As shown in Fig. 3(b), the performance of

Table 3: Test accuracy on CIFAR10 dataset under *attribute* test-time shift. All results are reported in mean(std error) format over three independent runs. $\Delta$ represents the difference between the accuracy on *original* test set and that on *attribute* test-time shift case.

| Method | Non-IID Dir(0.1) | | | Non-IID Dir(1) | | |
|---|---|---|---|---|---|---|
| | Original | Attribute Shift | $\Delta$ | Original | Attribute Shift | $\Delta$ |
| Local | 89.15(1.13) | 83.54(0.89) | -5.61 | 67.60(0.65) | 61.61(1.99) | **-5.99** |
| FedAvg | 64.25(1.42) | 50.09(1.36) | -14.16 | 73.23(0.38) | 57.17(0.34) | -16.06 |
| FedAvg-FT | 87.82(0.59) | 80.95(2.55) | -6.87 | 78.21(0.68) | 65.59(1.46) | -12.62 |
| FedProx | 59.79(5.91) | 52.25(2.53) | -7.54 | 72.02(0.37) | 61.72(1.29) | -10.3 |
| FedRep | 88.73(0.26) | 81.72(0.63) | -7.01 | 79.25(0.59) | 66.69(1.22) | -12.56 |
| FedBN | 83.09(4.56) | 78.14(0.57) | -4.95 | 77.67(1.07) | 66.90(1.31) | -10.77 |
| FedTHE | 90.55(0.41) | 81.91(0.54) | -8.64 | **80.49**(0.89) | 66.25(0.96) | -14.24 |
| Tent | **91.39**(0.77) | 85.72(1.24) | -5.67 | 78.61(0.63) | 71.50(1.29) | -7.11 |
| EATA | 91.37(0.78) | 85.60(1.16) | -5.77 | 79.72(0.62) | 70.44(1.67) | -7.28 |
| T3A | 90.79(0.73) | 85.91(1.68) | -4.88 | 78.80(0.95) | 70.93(0.68) | -7.87 |
| FedICON | 90.24(0.52) | **86.47**(1.27) | **-3.77** | 79.43(0.78) | **72.41**(1.29) | -7.02 |

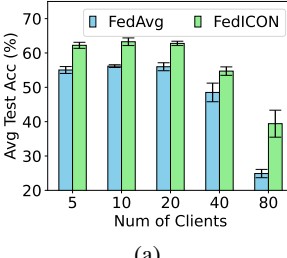
(a)

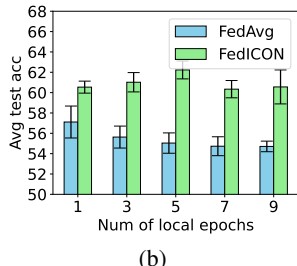
(b)

Figure 3: Test accuracy on Digit-5 under varying numbers of clients (*left*) and varying numbers of local epochs (*right*). Both are conducted in the covariate test-time shift cases. In (a), the number of clients ($m$) ranges from 5 to 80. The total number of samples is kept constant, so the number of samples in each client is fewer as $m$ goes larger. In (b), the number of local training epochs ranges from 1 to 9.

FedAvg keeps dropping as the number of local epochs increases, while FedICON partially enjoys the benefit of more local training, thus more communication-efficient than FedAvg.

**Extending to unsupervised scenarios.** The main framework of our proposed method is based on representation learning, which makes it flexible to migrate the whole learning paradigm into the unsupervised representation learning manner. During the local invariance extraction and global invariance sharing stages, we remove the label information. The feature encoder is purely trained in a self-supervised manner and the label information is accessible only when training the classifier. As is shown in Table 4, the extended version of our method FedICON, termed as FedICON-*us*, achieves the runner-up performance on Digit-5 and Office-10 under covariate test-time shift.

Table 4: Test accuracy on Digit-5 and Office-10 dataset under *covariate* test-time shift. The best and runner-up results are highlighted with **bold** and underline, respectively.

| Method | Digit-5 | Office-10 |
|---|---|---|
| Local | 34.00(3.12) | 58.07(2.84) |
| FedAvg | 55.04(1.00) | 57.55(1.46) |
| FedAvg-FT | 54.55(0.83) | 60.45(1.40) |
| FedICON | **62.23**(0.88) | **64.96**(1.43) |
| FedICON-*us* | 61.60(1.45) | 63.79(1.17) |

## 5.3 Ablation Studies

We carry out ablation studies to see the effect of each key technical component in FedICON. We alternatively remove the local invariance extraction, global invariance sharing and test-time adaptation procedure to see how these components affect the performance in handling test-time shift issues in FL. It is worth noting that when the local invariance extraction component is removed, the local training objective changes from the supervised contrastive loss to the conventional classification loss (*i.e.*, cross-entropy), and the global invariance sharing component is in charge of sharing the

Table 5: Comparison between the FedICON and its variants. Experiments are conducted on Digit-5 dataset under *covariate* test-time shift. The number of clients is 5. Each of them owns data samples from a specific domain. Corruptions are applied to their local test set to establish the *covariate* test-time shift setting. The number of communication rounds is 100.

| Design Component | Variants | | | | | | | |
|---|---|---|---|---|---|---|---|---|
| Local Invariance Extraction | | ✓ | | | ✓ | | ✓ | ✓ |
| Global Invariance Sharing | | | ✓ | | ✓ | ✓ | | ✓ |
| Test-Time Adaptation | | | | ✓ | | ✓ | ✓ | ✓ |
| Test Accuracy | 34.00 | 48.60 | 55.04 | 39.92 | 59.39 | 59.54 | 49.67 | 62.23 |

learnable parameters of the local model. In Table 5, we can observe that both global invariance sharing and test-time adaptation play an important role in improving the ability to deal with test-time shift problems. The effect of global invariance sharing is more notable because it is the core step to explore the inherent inter-client heterogeneity and broadcast the invariant encoding knowledge to benefit their local learning. The test-time adaptation brings about 1% to 5% improvement compared with the variants without this component, verifying the effectiveness of consistent representation alignment via a self-supervised learning scheme during the test phase.

## 6  Conclusion

In this paper, we aim to address the challenging but unexplored feature-level test-time shift problems in FL. We take the first initiative to thoroughly investigate the inherent *inter-client heterogeneity* in FL and propose to use it to deal with the test-time *intra-client heterogeneity*. We propose a novel algorithm, namely FedICON, where invariant information underlying heterogeneous data is extracted in a contrastive learning manner during the federated training phase and further used to deal with test-time shifts in a client. On our proposed benchmarks tailored for this problem, extensive experiments are conducted to show the superiority of FedICON compared with baseline methods.

This research has great potential in solving test-time shift problems in real-world FL systems. Providing theoretical guarantees for our work and discussion on privacy and security can be further explored in future studies. Considering the evident motivation, the novel setting and the effectiveness of our method, we believe this work is intriguing for future studies.

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
