# A Experimental Details

## A.1 Detailed Setting of Figure 1(b)

Experiments are implemented with FedAvg [63] on Digit-5 dataset [56] under feature shift non-IID setting [18]. The number of communication rounds is 100. The total number of clients is 4.

There are five subdatasets in Digit-5, *i.e.*, MNIST, SVHN, USPS, SynthDigits, and MNIST-M. For the Standalone case, clients own training samples from the same dataset of Digit-5, while for the Inter-Client Heterogeneity case, each client owns a specific dataset of Digit-5 as its training set. As for the test set, both of these two cases have to deal with test-time *domain* shifts where the test samples are from an unseen dataset. We follow the leave-one-out principle to select one of the five datasets as the test set in turn. The results report the mean accuracy across all experiments, and each experiment has three independent runs.

A two-layer CNN architecture is used as the backbone model here. The local training batch size is 32. An SGD optimizer with a weight decay rate 5e-4 and a learning rate 0.01 is used. The number of local epochs is 5.

## A.2 Detailed Setting of Test-Time Shifts

**Covariate shift.** The number of clients is 5, 4, and 6 for Digit-5, Office-10, and DomainNet, respectively. Compared with the feature-level distribution of the training set, we add covariate shift to the test set following [31, 65]. There is a hyperparameter ranging within $\{1, 2, 3, 4, 5\}$ that controls the severity level of corruption. We set the hyperparameter as the highest 5 for all covariate shift experiments.

**Attribute shift.** We followed the natural shift setting in [31]. There are 20 clients in total. Clients own the training set sampled from CIFAR10 and the test set sampled from CIFAR10.1 which is an attribute-shifted dataset of CIFAR10. The training set and test set of a specific client have the same label distribution, while the datasets across different clients are of various label distributions simulated by Dirichlet distribution. Similar to [31], we consider the hyperparameter $\alpha$ of Dirichlet distribution as 0.1 and 1, respectively.

**Domain shift.** The number of clients is 4, 3, and 5 for Digit-5, Office-10, and DomainNet, respectively. We follow the leave-one-out principle to select one domain as the test domain for all clients and the other domains as the training domains. Each client owns training data from a unique domain that is different from others.

## A.3 Detailed Setting of Baselines

The detailed setting and the hyperparameters of all the baselines are listed below.

- For **FedAvg** [63], the learning rate is 0.01, and the weight decay rate is 5e-4. The number of local epochs is 5. The number of communication rounds is 100 for Digit-5, Office-10, CIFAR10, and 20 for DomainNet.
- For **Local**, the hyperparameters are the same as **FedAvg** except that each client only performs local training without any communication.
- For **FedAvg-FT**, there is an additional local training round to update the global model based on the local training dataset.
- For **FedProx** [64], we tune the penalty constant $\mu$ from the limited candidate set $\{0.001, 0.01, 0.1, 1\}$ given in the original paper and select 0.01 for all settings.
- For **FedRep** [22], similar to the original paper, we use the last fully-connected layer as the local head and the other part of the model as the global representation layer.
- For **FedBN** [18], the batch normalization layers are kept locally trained while the other part of the model is globally shared. For simple CNN model architecture, we add batch normalization layers after the convolutional layers.
- For **FedTHE** [31], since there is no publicly available code at present, we follow the algorithm procedure claimed in the original paper and reproduce its framework using its default hyperparameters.

- For **Tent** [38], the step of test adaptation is set to be 1.
- For **EATA** [40], the number of samples used to compute the fisher information matrix and the trade-off between entropy and regularization loss are all set as the default value 2000. The entropy margin is set to be $0.4 \log(1000)$ and $\epsilon$ is set to be 0.05.
- For **T3A** [41], the size of support set is selected from $\{1, 5, 20, 50, 100\}$ and 50 is selected as the value for all settings.

## A.4 Detailed Setting of Model Architectures

For Digit-5, Office-10, and CIFAR10, we use a simple two-layer CNN as the backbone model, while for DomainNet dataset, we use ResNet18 pretrained on ImageNet1K as the backbone model.

The detailed architecture of the simple CNN is shown in Table 6.

Table 6: The two-layer simple CNN model architecture. FC refers to the fully connected layer.

| Layer | Details |
|---|---|
| 1 | Conv2D(3, 32, 5, 1), ReLU, MaxPool2D(2, 2) |
| 2 | Conv2D(32, 64, 5, 1), ReLU, MaxPool2D(2, 2) |
| 3 | FC(64 * 5 * 5, 64), ReLU |
| 4 | FC(64, 10) |

# B Additional Experiments

## B.1 The Effectiveness of FedICON on Handling Covariate Test-Time Shift Problem.

We also carry out experiments on Office-10 and DomainNet datasets to verify the effectiveness of FedICON on handling *covariate* test-time shift problem. The results are provided in Table 7 and Table 8, respectively. The results demonstrate that FedICON consistently outperforms existing baselines during the local test of all clients. We additionally show the experimental results of Digit-5 under *domain* test-time shift. The results are provided in Table 9. The results demonstrate that FedICON outperforms the baselines in most clients.

Table 7: Test accuracy on Office-10 dataset under *covariate* test-time shift. All results are reported in mean(std error) format over three independent runs.

| Method | Amazon | Caltech | DSLR | Webcam |
|---|---|---|---|---|
| Local | 60.94(1.56) | 39.70(2.45) | 66.67(4.77) | 64.97(2.59) |
| FedAvg | 61.63(0.80) | 35.41(1.43) | 67.71(1.80) | 65.45(1.81) |
| FedAvg-FT | 64.93(0.80) | 41.04(0.93) | 67.79(2.43) | 68.04(1.45) |
| FedProx | 61.81(1.83) | 36.15(1.36) | 64.58(1.80) | 67.23(3.53) |
| FedRep | 55.90(2.57) | 32.15(3.59) | 44.79(9.02) | 54.07(4.17) |
| FedBN | 49.83(3.01) | 38.37(1.12) | 64.58(3.61) | 56.50(5.95) |
| FedTHE | 62.85(0.80) | 37.33(2.35) | 63.54(3.61) | 69.49(3.39) |
| Tent | 58.51(3.94) | 44.44(1.33) | 61.46(7.22) | 67.23(4.27) |
| EATA | 58.16(4.24) | 44.30(1.43) | 61.46(7.22) | 65.54(2.59) |
| T3A | 58.68(2.67) | 42.67(2.31) | 63.54(3.61) | 66.10(1.69) |
| FedICON | **71.53**(0.30) | **48.44**(0.89) | **69.79**(6.51) | **70.06**(3.53) |

Table 8: Test accuracy on DomainNet dataset under *covariate* test-time shift. All results are reported in mean(std error) format over three independent runs.

| Method | Clipart | Infograph | Painting | Quickdraw | Real | Sketch |
|--------|---------|-----------|----------|-----------|------|--------|
| Local | 72.05(1.88) | 36.53(1.72) | 74.31(0.23) | 50.20(0.57) | 79.25(1.34) | 67.78(0.89) |
| FedAvg | 78.94(0.94) | 37.32(0.54) | 70.60(0.23) | 50.05(1.48) | 77.90(0.23) | 70.13(1.15) |
| FedAvg-FT | 77.23(1.75) | 37.91(0.11) | 75.27(0.34) | 46.80(1.84) | 80.44(0.46) | 70.76(0.51) |
| FedProx | 79.13(0.67) | 37.95(0.32) | 69.47(0.23) | 49.95(2.62) | 77.77(0.64) | 69.86(0.26) |
| FedRep | 73.67(1.48) | 36.30(2.26) | 74.47(1.60) | 53.70(0.99) | 79.05(2.21) | 67.78(0.64) |
| FedBN | 78.32(0.94) | 37.15(0.75) | 74.82(2.40) | 44.75(4.60) | 80.32(0.41) | 70.04(1.02) |
| FedTHE | 66.63(0.13) | 35.01(0.43) | 60.58(0.23) | 43.70(1.13) | 56.74(0.64) | 62.18(0.64) |
| Tent | 49.24(0.54) | 27.47(0.11) | 43.62(1.37) | 30.30(0.42) | 31.18(0.17) | 46.21(0.51) |
| EATA | 49.43(0.54) | 27.09(0.22) | 42.89(1.71) | 29.35(0.07) | 30.90(0.23) | 46.21(0.51) |
| T3A | 78.71(1.61) | 36.77(0.65) | 74.76(1.94) | 53.10(7.21) | 80.32(0.64) | 72.65(1.40) |
| FedICON | **79.66**(0.27) | **38.43**(0.75) | **76.01**(0.11) | **57.40**(0.71) | **84.72**(0.23) | **76.44**(2.17) |

Table 9: Test accuracy on Digit-5 dataset under *domain* test-time shift. All results are reported in mean(std error) format over three independent runs.

| Method | Test Domain | | | | |
|--------|-------------|------|------|-------|---------|
| | MNIST | SVHN | USPS | Synth | MNIST-M |
| Local | 41.35(2.60) | 14.98(0.48) | 32.67(2.37) | 18.68(0.53) | 18.22(0.94) |
| FedAvg | 81.30(1.81) | 21.04(1.21) | 54.32(0.67) | 42.04(1.65) | 31.92(0.67) |
| FedAvg-FT | 79.63(1.95) | 20.86(0.93) | 53.79(1.07) | 40.98(1.91) | 31.44(0.67) |
| FedProx | 83.28(1.02) | 23.26(1.49) | 53.87(1.16) | 41.75(1.02) | 31.96(0.89) |
| FedRep | 57.39(3.88) | 17.72(1.03) | 42.72(1.56) | 24.08(1.23) | 20.88(0.20) |
| FedBN | 78.65(1.63) | 20.74(1.86) | 63.41(3.67) | 37.04(1.75) | 23.20(0.43) |
| FedTHE | 79.09(1.30) | 20.78(1.55) | 55.62(0.47) | 41.98(1.10) | 30.74(0.95) |
| Tent | 74.65(2.41) | 32.89(0.45) | 65.61(0.16) | 45.62(1.19) | 28.36(0.87) |
| EATA | 74.91(1.77) | **33.02**(0.40) | 65.99(0.16) | 45.71(1.08) | 28.29(0.56) |
| T3A | 79.30(0.80) | 19.93(0.82) | 53.86(1.57) | 41.10(1.12) | 31.22(0.37) |
| FedICON | **89.75**(0.31) | 30.22(2.56) | **74.14**(0.40) | **51.73**(1.60) | **36.03**(1.01) |

## B.2  The Learned Local and Global Invariance.

In Table 10, we report the average standard deviation of representations within the same class in a client and across clients, respectively. The experiments are run on Digit-5 dataset under covariate test-time shift.

A lower standard deviation means that the representations within a class are much more similar. It can be seen that the representation variance of FedICON is lower than those in other methods, illustrating the ability of FedICON to extract and share invariance, hence alleviating the test-shift problem.

Table 10: Average standard deviation of representations within the same class in a client and across clients.

| Method | In A Client | Across Clients |
|--------|-------------|----------------|
| Local | 0.036 | 0.079 |
| FedAvg | 0.045 | 0.064 |
| FedRep | 0.041 | 0.065 |
| FedICON | **0.024** | **0.058** |

# C   Algorithm

The algorithm of FedICON is presented in Algorithm 1.

---

**Algorithm 1 FedICON**

---

**Training Phase** with **Input:** Training dataset $\mathcal{D}_i^P$, parameter of feature encoder $\theta_i$, $i = 1, \cdots, m$.
**Server executes:**

  1: Initialize feature encoder with parameter $\bar{\theta}$.
  2: **for** each round $T = 1, 2, ...$ **do**
  3:      **for** each client $i$ **in parallel do**
  4:         $\theta_i \leftarrow \text{LocalUpdate}(i, \bar{\theta})$
  5:      **end for**
  6:      Update global feature encoder by Eq. (6).
  7: **end for**

**LocalUpdate**$(i, \bar{\theta})$:

  1: **for** each local epoch **do**
  2:      **for** each batch in $\mathcal{D}_i$ **do**
  3:         Compute $L_{i,\text{sup}}$ and update $\theta_i$ by Eq. (4).
  4:      **end for**
  5: **end for**
  6: **return** $\theta_i$

**Test Phase at the $i$-th Client** with **Input:** Test dataset $\mathcal{D}_i^Q$, parameter of feature encoder $\theta_i$.
**LocalTest**$(i, \theta_i)$:

  1: **for** each local epoch **do**
  2:      **for** each batch in $\mathcal{D}_i^Q$ **do**
  3:         Compute $L_{i,\text{us}}$ and fine-tune $\theta_i$ by Eq. (7).
  4:      **end for**
  5: **end for**
  6: **return** $\theta_i$

---