# OpenReview forum: "Is Heterogeneity Notorious? Taming Heterogeneity to Handle Test-Time Shift in Federated Learning"
_NeurIPS.cc/2023/Conference — NeurIPS 2023 poster_

### Official Review · Reviewer_v7Ae · 2023-06-27

**Soundness:** 2 fair
**Presentation:** 3 good
**Contribution:** 2 fair
**Rating:** 4
**Confidence:** 4

**Summary:**

The authors propose a scheme to handle the test-time shift at each FL client. To achieve this, during training, a contrastive learning is adopted to extract invariant information within the same class (class-level invariant information) at each client. The updated models at the clients are also aggregated via FedAvg. During testing, at each client, the model is further fine-tuned using the test dataset and the corresponding unsupervised loss function.


**Strengths:**

1. The paper is generally well written and easy to understand. The figures also well describe the high-level concepts of the proposed methods.

2. This paper proposes strategies for both training and testing during FL, to handle the test-time shift issue, which is practical.

3. The authors consider various datasets for experiments including DomainNet.


**Weaknesses:**

1. Section 4.1.1: First of all, in 4.1.1, the authors are trying to extract invariant information within the same class, which shares the same concept with domain generalization that learns domain-invariant features during training. However, comparison with these methods are lacking. For example, one can apply any domain generalization schemes for local training to learn domain-invariant features, and then aggregate the models according to 4.1.2. Note that in domain generalization literatures, “domain shift” generally includes covariate shift, attribute shift, and domain shift the authors mentioned in this paper. Existing domain generalization methods can generally handle all these shifts.

2. Related to the comment above, there are many works that focus on adopting contrastive loss for domain generalization. It is not clear what is the authors’ novelty compared to the existing works on contrastive learning loss in Section 4.1.1

3. Section 4.1.2: This part is just about FedAvg. Hence, in my point of view, Section 4.1 is about performing updates locally to extract invariant information, and then aggregating the models. However, I do not find the novelty compared to applying existing works in Section 4.1

4. Section 4.2: During testing, a test dataset needs to be available in each client for fine-tuning. This may not be practical when test samples arrive one-by-one. Moreover, focusing on a specific client (and given the well-trained models), what is the novelty of Section 4.2 compared to other test-time adaptation methods?

5. Experiments: For experiments, as mentioned in my first comment above, the important baselines are missing. Especially, the authors’ scheme is the only method that simultaneously adopts (i) generalization during training and (ii) test-time adaptation at inference. To validate the effectiveness of the idea in Section 4.1.1, the authors need to consider schemes that apply domain generalization methods instead of Section 4.1.1 and combine it with Section 4.2, and compare it with FedICON. Similarly, to validate the effectiveness of the idea in Section 4.2, the authors may combine other test-time methods (e.g., Tent, T3A) with the idea in Section 4.1.1, and compare the scheme with FedICON.

6. Experimental setup: It is not clear how data are distributed across clients. Moreover, during testing, does the proposed method assumes that full test dataset is accessible at each client to finetune the model?

Overall, this paper can be viewed as a combination of domain generalization learning (or contrastive learning) and test-time adaptation in FL, but I find the novelty of the paper to be relatively weak, since there are no clear descriptions showing the difference compared to various existing works and there are no experiments comparing with the corresponding baselines.

**Questions:**

Another question is that, what happens if the classifier is not personalized but shared by aggregation?

Moreover, what is the experimental setup of Fig. 1(b) showing the advantage of heterogeneous data compared to homogeneous setup?

Regarding other questions, please refer to the weakness above.


**Limitations:**

Yes.

---

> ### Author Rebuttal · Authors · 2023-08-10
>
> Thanks for all the valuable comments and questions.
>
> **Comparison with the combination of DG and FL.** We used three DG-originated test-time adaptation (TTA) methods as baselines because they are more aligned with our setting and can be easily implemented in an FL framework. Note that not all existing DG methods can fit into FL settings and solve the test-time shift problem appropriately.
>
> **Concept confusion issue.** We would like to explain that the concept of “domain shift” in our paper refers to the more general cases of feature-level shifts compared with the covariate and attribute shifts, which can be seen as special cases of the general domain shift. In general domain shift cases, the training and test data are from different sub-datasets of the multi-domain datasets. We will add a clarification in the final version. We take the first step to comprehensively investigate this problem in FL.
>
> **Limited novelty compared with existing DG methods using contrastive loss.** We have unique novelty compared with existing contrastive learning-based DG methods in terms of **target problem** and **technical design**. *First*, we focus on the test-time shift problem in FL rather than the DG problem where the source domain and the target domain are usually clearly pinpointed. There are some unique challenges to solve in FL, e.g., how local training is conducted, how the knowledge across clients should be aligned, and how privacy is guaranteed. *Second*, existing works [1,2] usually use contrastive learning as a regularization term to the original cross-entropy loss instead of using a pure contrastive learning-based framework to optimize the model so as to resolve the test-time shift or domain shift issues. But in our method, we establish the framework of FedICON based on the representation learning scheme. We focus more on extracting and sharing the invariance knowledge by contrastive learning in FL setting.
>
> [1] Self-supervised Contrastive Regularization for Domain Generalization. In ICCV 2021.
>
> [2] Proxy-based contrastive learning for domain generalization. In CVPR 2022.
>
> **Novelty of Sec 4.1.** Compared to existing works, Sec 4.1 has the following novelty. First, the model architectures are different. We utilize the representation learning framework to solve the problem. While most FL methods train the whole model and do not take the intermediate output into consideration. Second, the loss functions are different. Most existing FL methods utilize cross-entropy as their loss functions while our method computes the loss based on the output of the feature encoder in a contrastive learning manner. Third, the communication protocols are different. In FL, some methods average all the learnable parameters at the server, while others may average part of the model parameters to achieve a specific goal, e.g., personalization. The selection and design of the communication protocol is an important component of the FL methodology. It is non-trivial to specify the concrete global update/sharing strategy in the proposed method.
>
> **Test samples arrive one-by-one.** Please refer to **C1** in the general response.
>
> **Novelty of Sec 4.2 compared to other TTA methods.** Sec 4.2 is proposed to keep the losses of the training and test phase unified and make the whole process consistent. To the best of our knowledge, we have not seen any existing works solving the test-time shift problem in this way, not to mention formulating the problem and developing a solution for federated frameworks.
>
> **Missing baselines.** We have considered three DG-originated test-time adaptation (TTA) methods and implemented them in the vanilla federated learning framework to validate the performance of our method. However, it is hard to directly combine TTA methods with Sec 4.1.1 because most of them (e.g., Tent, EATA, T3A) require the output of the classifier head which is not involved in Sec 4.1.1. Also, it is hard to directly combine Sec 4.2 with DG methods, because Sec 4.2 is kept in a unified form with the loss in Sec 4.1.1. If the training loss is not in a pure contrastive learning manner, the operation in Sec 4.2 may lead to severe performance degradation or divergence. Note that we focus on test-time shift problem in FL and compare with ten baselines that can fit into our setting. We will add more discussion on the selection of baselines in the final version.
>
> **Data setup details.** Since we followed the benchmark setting in [1,2] as mentioned in the paper, we just illustrate how the test-time shifts are simulated and the values of hyperparameters in Appendix A.2. We will more details in the final version to make the data setup clearer.
>
> [1] Federated learning on non-iid features via local batch normalization. In ICLR 2021.
>
> [2] Test-time robust personalization for federated learning. In ICLR 2023.
>
> **Full test set requirement.** Our method does not require access to the full test dataset. Please refer to **C1** in the general response for detail.
>
> **The case where classifier is not personalized but shared.** We compared several variants for classifier training on top of the frozen feature encoder and selected the case achieving the best performance. We chose not to present the results in the main text to 1) help isolate the effects of the proposed key components and the classifier part; 2) emphasize that our method is open to different kinds of additional classifier training strategies according to the specific data setting. We add the experimental results to Table 1 in the uploaded file and will provide a sufficient illustration in the final version. We hope the new results can efficiently address your concerns about the classifier alignment.
>
> **Detailed Setting of Figure 1(b).** We provided the detailed specification of the setting of Fig. 1(b) in Appendix A.1 titled “Detailed Setting of Figure 1(b)” as shown below. We will add a brief specification in the main text in the final version.

---

> > ### Comment · Reviewer_v7Ae · 2023-08-11
> >
> > I would like to appreciate the authors for the detailed response. At the same time, I am sorry to say that I am still leaning against rejection, especially due to the experiments. Although the authors mention that they focus on test-time shift instead of DG, they both share some similar philosophy in finding the invariant knowledge during training to gain robustness during testing. And if my understanding is correct, the authors are actually doing this in Sec. 4.1.  More specifically, the authors are (i) extracting invariant information during training and (ii) conducting test-time adaptation at inference.
> >
> > To reiterate my comment, the authors’ scheme is the only method that simultaneously adopts (i) generalization during training and (ii) test-time adaptation at inference. And thank you for letting me know that Sec. 4.1 and Sec. 4.2 are not compatible with other schemes.
> >
> > However, the baselines are currently only focusing either training or testing. For example, Tent, T3A only focus on the testing-phase. Can their performance get improved by adopting DG methods during training, to better learn invariant features/characteristics? (There are many DG methods that are applicable to local updates for FL clients to learn invariant knowledge). At the same time, the 6 baselines on FL only focus on the training stage. Can their performance get improved by doing test-time adaptation during inference? A natural question that people might be interested in is, can existing combinations perform better? If not, what makes the authors’ scheme to perform the best? I believe these results as well as the comprehensive discussions should have been included in the original submission. Without these comprehensive analysis, I feel that the current results are less surprising, especially considering the bar of NeurIPS.

---

> > > ### Author Response · Authors · 2023-08-12
> > >
> > > We sincerely thank you for the prompt reply. We would like to clarify that we focus on the test-time shift problem in the **federated learning** setting. It is essential and reasonable to compare the proposed method with existing typical generic FL and personalized FL algorithms, which perform better on either generalized or personalized data distribution and can be implemented as a direct solution for our target problem. We admit there are many possible combinations among 1) the training-phase DG methods, 2) the test-phase TTA methods, and 3) different generic FL and personalized FL methods. However, it is non-trivial to develop an algorithm that not only fits the test-time shift scenarios in FL but also outperforms the existing state-of-the-art method, FedTHE [3], on various test-time shift settings. By providing a new state-of-the-art solution, we sincerely hope our paper can shed light on exploring the utilization of unique properties of FL systems compared with centralized systems, e.g., the inherent data heterogeneity across clients, to help alleviate the practical test-time shift problems in FL.
> > >
> > > [1] In search of lost domain generalization. In ICLR 2021.
> > >
> > > [2] Test-Time Classiﬁer Adjustment Module for Model-Agnostic Domain Generalization. In NeurIPS 2021.
> > >
> > > [3] Test-time robust personalization for federated learning. In ICLR 2023.

---

> > > > ### Comment · Reviewer_v7Ae · 2023-08-15
> > > >
> > > > Thanks for the response. I went through the paper and the related works again, and I am still not convinced. As the authors mentioned, since the focus of this paper is on the relatively new setup with "test-time shift" + "federated learning", it is very obvious that existing works generally do not perform well. And, it is very natural to ask the following question: Can this problem be handled based on the combination of existing literature? The authors currently do not provide answer to this, as I described above.
> > > >
> > > > Especially, it is important to note that existing test-time adaptation methods are generally applicable when any pre-trained model is given. However, for the authors' work, it is necessary to do something during training, and this part shares some common philosophy with DG or learning invariant knowledge. Hence, I do not view this work as a pure test-time adaption strategy (because it is not applicable when simply a pre-trained model is given), but as a joint training + test-time strategy (although the goal is still test-time adaptation, it is required to do something during training). This again will make the readers curious about the above question. This should be also written clearly throughout the intro and related works.
> > > >
> > > > Due to the above reasons, I believe new experimental results as well as comprehensive discussions and additional writing are required.

---

### Official Review · Reviewer_pJGu · 2023-07-04

**Soundness:** 2 fair
**Presentation:** 3 good
**Contribution:** 2 fair
**Rating:** 4
**Confidence:** 4

**Summary:**

This paper propose FedICON, which uses inter-client heterogeneity to handle intra-client heterogeneity. During training, FedICON uses contrastive learning locally to extract invariant class-conditional information, and performs global invariance sharing under inter-client heterogeneity. During testing, the feature extractor is adapted with contrastive learning. Extensive experiments validate the effectiveness of FedICON.

**Strengths:**

- The paper makes clear definitions of inter-client and intra-client heterogeneities. Meanwhile, it is insightful to propose that inter-client heterogeneity in FL can be used to tackle intra-client heterogeneity.
- FedICON has good performance on a variety of experimental settings. Especially, the author tried different types of heterogeneities.

**Weaknesses:**

- Global invariance sharing. It is unclear why sharing the model parameter of feature extractor can achieve global invariance sharing. Since FedICON does not compare representations from different clients, FedICON only guarantee local invariance on each client, but not global invariance. For example, consider a binary classification problem, a feature extractor might map positive/negative examples on client 0 to (0, 1), (0, 3), and map positive/negative examples on client 1 to (0, -2), (0, -4). In this way, although all positive samples on client 0 are mapped to the same representation (0, 1), positive samples on client 0 and 1 are mapped to different representations (0, 1) and (0, -2).
- Since this paper focuses on how inter-client heterogeneity can help tackling intra-client heterogeneity, I believe the author may consider emphasizing the type of distribution shifts for each heterogeneity in the main text, instead of in the supplementary material.

**Questions:**

- Could you provide more information about whether FedICON learns local and global invariances? E.g., by numerical results of the variance of representations, or visualization (not required since it could be hard during rebuttal)
- Other questions that I am interested:
  - In line 190, the authors mentioned that they generate two random augmenteed views for each sample. Usually this is for constructing positive pairs for unlabelled data (e.g., in self-supervised learning). However here the setting is supervised. Also in equation (3) and (4) it seems the augmented data is not fed into the feature encoder. I am confused whether the augmentation is used during training.

**Limitations:**

The limitations are clearly stated.

---

> ### Author Rebuttal · Authors · 2023-08-10
>
> **The reason why global invariance sharing can work.** Thanks for the valuable comments. There might be some misunderstanding towards the global invariance sharing part. There is no actual global invariance extracted among participating clients. The shared global invariance in the paper refers to the invariance encoding ability underlying model parameters. We claim that sharing the model parameter of feature extractor can achieve global invariance sharing because the parameter sharing process broadcasts the invariant encoding knowledge acquired during local training and hence mutually boost invariance extraction globally. As for the binary classification example, after aligning the model parameters of client 0 and client 1, it is of great probability that the positive samples on them are mapped to the same representation, e.g., (0, -0.5) even if there is a certain degree of deviation between their data.
>
> **Emphasizing heterogeneity types in the main text instead of supplementary material.** Thanks for the valuable comments. To make it convenient for the readers to capture key information about the heterogeneity setup, we will move the specification of several distribution shifts (inter- and intra-heterogeneity) to the main text in the final version. Due to the space limit, we still leave some detailed specifications of each feature-level shift, i.e., the number of clients, the reference previous works, and the hyperparameters for data setup, to Appendix A.2 but we provide a summary of them in Sec 5.1 to increase the readability.
>
> **More information about the learned local and global invariance.** Thanks for the valuable comments and questions. In Table 5 of Sec 5.3, we show performance improvement achieved by local invariance extraction and global invariance sharing component. Furthermore, we show how the local and global invariance are learned in the following.
>
> In the Table below, we report the average standard deviation of representations within the same class in a client and across client, respectively. The experiments are run on Digit-5 dataset under covariate test-time shift.
>
> A lower standard deviation means that the representations within a class are much more similar. It can be seen that the representation variance of FedICON is lower than those in other methods, illustrating the ability of FedICON to extract and share invariance and hence alleviate the test-shift problem.
>
> |Method|In A Client|Across Clients|
> |-|-|-|
> |Local|0.036|0.079|
> |FedAvg|0.045|0.064|
> |FedRep|0.041|0.065|
> |FedICON(Ours)|0.024|0.058|
>
> We hope these new results can efficiently address your concerns about whether FedICON extracts local invariance and acquires global invariance encoding ability in our proposed method. We will add the complete results in the final version. Also, it would be beneficial to use some visualization techniques to present this point in a more indicative manner. We will spend some time on visualization and present the results in the final version.
>
> **Whether the augmentation is used during training.** Thanks for the valuable comments and questions. As mentioned in line 190, given an input batch of data, we ﬁrst apply data augmentation to obtain two random augmented views of each sample, which doubles the size of the input dataset. Then, for each sample $\mathbf{x}$, we conduct Eq. 3 to obtain its feature embedding and compute the supervised contrastive loss in Eq. 4. It should be noted that both the original data and the augmented data are involved in $P(\mathbf{x})$ and $A(\mathbf{x})$ and fed into the feature encoder so that they contribute to the model training process.
>
> Some previous works also construct positive pairs for contrastive learning in the supervised setting [1,2,3], which ensures the existence of at least one positive sample within the input data batch when contrasting the anchor $\mathbf{x}$. We will add more explanations to make it clearer in the revised version.
>
> [1] Supervised contrastive learning. In NeurIPS 2020.
>
> [2] Targeted supervised contrastive learning for long-tailed recognition. In CVPR 2022.
>
> [3] Federated learning from pre-trained models: a contrastive learning approach. In NeurIPS 2022.

---

> > ### Comment · Reviewer_pJGu · 2023-08-15
> > **Response to Authors**
> >
> > Thanks a lot for your rebuttal. Your experiments regarding the learned local and global invariance show that your proposed FedICON can reduce both the intra-client and inter-client variance, which answers my question 1, thanks!
> >
> > After reading your rebuttal and re-reading the paper, I am still very confused regarding my question of weakness 1. As stated in the abstract (line 10), the purpose of this paper is to use “inter-client heterogeneity in handling intra-client heterogeneity”. However, in the algorithm design,
> > - In 4.1.1, each client’s local training extract local invariance. I believe it is expected that the representation should be “invariant” to (1) augmentation and (2) intra-class natural variance.
> > - In 4.1.2, the parameter of feature extractor is shared across clients. I believe it is only expected that the representation will be “invariant” to the aforementioned (1) and (2) on all clients, in other words, the union of random augmentations and intra-class natural variance.
> >
> > I am very confused how inter-client heterogeneity is exploited in the algorithm. It seems to me that FedICON only use the union of intra-client variance. The only module in FedICON regarding multiple client is the feature extractor parameter averaging, which is also used in FedAvg (although FedAvg averages all the parameters). The statement of “it is of great probability that the positive samples on them are mapped to the same representation, e.g., (0, -0.5) even if there is a certain degree of deviation between their data” lacks evidence, and I highly disagree with it: if simply sharing parameter can achieve invariance, what is the purpose of domain adaptation and generalization algorithms?
> >
> > I recognize that FedICON can achieve great performance across datasets and distribution shifts. However, I believe it is also important to figure out “why FedICON works”, and whether it matches with your motivation of using inter-client heterogeneity. Here are some suggestions for ablation study:
> > 1. Is inter-client heterogeneity really exploited? You may consider running experiments with FedICON under both IID partition and non-IID partition, training to a same accuracy (to avoid optimization challenge under non-IIDness), and testing on shifted data. If inter-client heterogeneity is really exploited, FedICON under non-IID client distribution should have higher accuracy.
> > 2. Is the high performance a result of data augmentation? I recognize that data augmentation is widely used. However, we know that using data augmentation during training can obviously improve the robustness of model to test-time shifts (especially when augmentation and distribution shifts share many similarities). When comparing FedICON to baselines, I notice that baseline methods do not use data augmentation (correct me if I am wrong), which is unfair to baselines since augmentation can be easily incorporated into them. It may be more meaningful if you can (1) adding data augmentation to baselines, or (2) removing data augmentation from FedICON. I believe it will be a fairer comparison.
> >
> > I apologize for proposing new experiments during the discussion. However I do believe that it is very important to test whether the proposed algorithm matches with you motivation. You don’t need to follow my suggestion exactly, as long as my question of “whether and how inter-client heterogeneity is exploited” is answered.

---

> > > ### Author Response · Authors · 2023-08-16
> > >
> > > We sincerely thank you for the reply. We are glad to hear that part of your concern about the simultaneous learned local and global invariance is alleviated. As for your further concern about how inter-client heterogeneity is exploited to solve the intra-client heterogeneity problem, we hope the following explanation and experimental results can help address it.
> > >
> > > **Whether the inter-client heterogeneity is really exploited.** We agree with your suggestion on running experiments under both IID and non-IID partitions to verify the positive effect of inter-client heterogeneity. In fact, we have conducted motivated experiments that consider both the IID and non-IID partitions in **Fig. 1(b)**. **Standalone** refers to the case where each client owns the same dataset of Digit-5 (IID partition), while **Inter-Client Heter** refers to the case where each client owns a different dataset from Digit-5 (non-IID partition). Usually, it is hard to obtain the same training accuracy under different data settings because of the objective inconsistency under different data settings [1]. The fairer way is to fix the number of communication rounds and compare their accuracy on the shifted test set. *The results show that inter-client heterogeneity can truly improve the ability of clients to handle test-time shift problems compared to training on homogeneous data.* Based on this observation, we further developed our method FedICON to fully take advantage of the benefit brought by inter-client heterogeneity during training time, and then, at the test phase, tune the local model of each client on its private test data to conquer the intra-client heterogeneity.
> > >
> > > [1] Tackling the Objective Inconsistency Problem in Heterogeneous Federated Optimization. In NeurIPS 2020.
> > >
> > > **The effect of data augmentation.** The performance improvement benefits from the data augmentation but is not entirely due to it. We did not perform an independent ablation study on the data augmentation component because we view it as a part of the local invariance extraction. However, in our ablation study (Table 5), we consider a variant that removes the data augmentation module. That variant replaces the whole local invariance extraction model by conventional supervised learning module (cross-entropy loss), the accuracy drops from 62.23% to 59.54%, but still higher than baselines where the highest average accuracy is 59.14% (achieved by Tent). According to your suggestion, we provide the following tables under *covariate* test-time shift to explicitly show the independent effect of data augmentation. The upper table reports the average accuracy in Table 1 of the main text and the bottom table reports supplementary results to Table 5 of the main text, suggesting that the performance improvement only partially results from the data augmentation module.
> > >
> > > |**Method**|Local|FedAvg|FedAvg-FT|FedProx|FedRep|FedBN|FedTHE|Tent|EATA|T3A|FedICON(Ours)|
> > > |-|:-:|:-:|:-:|:-:|:-:|:-:|:-:|:-:|:-:|:-:|:-:|
> > > |**Avg Acc**|34.00|55.04|54.55|55.23|38.14|54.80|45.93|*59.14*|58.88|51.60|**62.23**|
> > >
> > > |**Component**|||  **Variants**|||
> > > |-|:-:|:-:|:-:|:-:|:-:|
> > > |Local Invariance Extraction - Data Augmentation|&check;|&check;|||
> > > |Local Invariance Extraction - Local Contrastive Loss|&check;||&check;||
> > > |**Avg Acc**|62.23|60.87|61.64|59.54|
> > >
> > > We hope the existing ablation study and the above additional results can well address your concern on the effect of the data augmentation module.

---

> ### Author Response · Authors · 2023-08-18
>
> Dear Reviewer pJGu,
>
> We sincerely appreciate the feedback and suggestions you have provided to our paper. We have tried our best to address the concerns raised by you. We sincerely hope our responses have solved your concerns and we remain fully available to respond to any extra inquiries that may arise during the discussion phase.
>
> Best Regards,
>
> Authors

---

> > ### Comment · Reviewer_pJGu · 2023-08-21
> > **Thanks for your response**
> >
> > Thanks a lot for your response. I apologize for late response since I am seriously ill recently.
> >
> > Regarding the effect of data augmentation, I think the additional experiment answers my question. Thanks!
> >
> > Regarding the whether the inter-client heterogeneity is really exploited in FedICON, I still believe that experiments on FedICON, with the setting in Table 1, while changing IID paritition to non-IID paritition, is important. I understand the motivation behind Fig 1(b), however, the algorithm used here seems not to be FedICON.

---

> > > ### Author Response · Authors · 2023-08-21
> > >
> > > Dear Reviewer pJGu,
> > >
> > > We are sorry to hear that you are ill and hope you to be better soon. We again really appreciate your feedback. We are sorry for not showing the results of FedICON under both IID and non-IID partition cases. What we try to verify is the generally existed phenomenon that inter-client heterogeneity can help alleviate the test-time shift problem. In fact, we have conducted the experiments of FedICON under IID and non-IID partition cases but omitted the results in the motivated experiments. We will follow your suggestion and add the following results in the format of bar chart to the experimental part. We do believe this will address the readers' concern about whether inter-client heterogeneity is exploited in FedICON.
> > >
> > > |**Partition**|	**Method**|	mn|	sv|	ys|	syn|	mm|
> > > |-|-|-|-|-|-|-|
> > > |IID	|FedAvg|	77.18,8.15|	15.30,3.30|	42.35,4.36|	18.47,2.45|	21.00,1.92|
> > > |IID   | FedICON|84.00,0.88|18.45,0.17|47.08,2.12|29.55,2.23|28.66,1.84|
> > > |non-IID| FedAvg|	86.18,0.72|	38.60,1.84|	58.19,2.78|	51.50,1.13|	40.72,1.15|
> > > |non-IID|FedICON|	89.67,1.48|	45.23,0.35|	72.13,2.22|	56.13,0.63|	47.98,0.50|

---

### Official Review · Reviewer_CniS · 2023-07-06

**Soundness:** 4 excellent
**Presentation:** 4 excellent
**Contribution:** 4 excellent
**Rating:** 8
**Confidence:** 4

**Summary:**

To deal with the feature-level test-time shift problem in federated learning, this paper proposes to leverage the inherent heterogeneity across clients based on a contrastive learning method, named FedICON. Clients acquire invariance encoding ability on heterogeneous source data and further boost the performance in the test phase. Experiments on various datasets show the effectiveness of FedICON compared with baseline methods.

**Strengths:**

1. It is a novel idea to leverage heterogeneous properties underlying FL systems to deal with the commonly existing test-time shift problem within a client. Compared with centralized test-time adaptation studies, the heterogeneous problem in the training phase is a characteristic of FL.
2. Sufficient experiments are conducted and the main results show that FedICON has achieved improvement for almost all participating clients under various feature-level shifts.
3. The paper is easy to follow and well-organized. The authors provide clear and novel definitions of inter-client and intra-client heterogeneity. The motivation for utilizing inter-client heterogeneity to deal with intra-client heterogeneity is well demonstrated.

**Weaknesses:**

1. The concepts of test-time shift and intra-client heterogeneity seem to refer to the same thing, which is not well clarified in the main paper.
2. It is better to give a better explanation of the fundamental difference between label-level and feature-level test-time shifts, which makes it easier for the readers to understand why the shifts should be studied separately.

**Questions:**

1. Is it possible to deploy the proposed method in an online manner, e.g., the online test deployment in [1], to handle the dramatic shifts during the test phase?
[1] FedTHE: Test-Time Robust Personalization for Federated Learning. In ICLR 2023.

2. Given a local model, how to choose an appropriate feature space for contrastive learning?

**Limitations:**

Yes. The authors have addressed the limitations.

---

> ### Author Rebuttal · Authors · 2023-08-10
>
> **Difference between test-time shift and intra-client heterogeneity.** Thanks for the valuable comments. The intra-client heterogeneity we defined in this paper refers to the test-time shift issue, especially in federated learning. The concept of test-time shift is commonly used in general machine learning, e.g., both centralized and decentralized frameworks. Since test-time shift is a more well-known challenging issue recognized by the community, we use it to pinpoint the core challenge addressed by this paper. However, to better formulate the problem of test-time shift in FL and make the notation more unified, we propose the concept of intra-client heterogeneity existing in the inference stage of a specific client, as a counterpart of inter-client heterogeneity in the training stage among all the participating clients. We will add more clarification on the concept of test-time shift and intra-client heterogeneity in the revised version.
>
> **Difference between label-level and feature-level test-time shifts.** Thanks for the valuable comments. During the federated training stage, label-level and feature-level shift across clients are two different kinds of statistical heterogeneous issues for federated learning, which occurs in the output space and input space, respectively. Usually, different strategies are designed to solve these two issues. For example, to alleviate the feature-level shift issue, the feature encoder is kept personalized or finetuned at each client; to alleviate the label-level shift issue, the classifier is tuned instead. As for the test-time shift, feature-level and label-level shift are still two different challenges that deserve specific technical design to alleviate them respectively. Experimental results in [1] show that, even though a method that is effective for most feature-level shift cases may still not work well for the label-level shift case. Hence, it is better to discuss these two shift cases respectively. We will add more discussion on the explanation of the fundamental difference between these two kinds of test-time shifts in the revised version.
>
> [1] Test-time robust personalization for federated learning. In ICLR 2023.
>
> **Deployment in an online manner.** Thanks for the valuable question. In our current method, according to Eq. 7, the size of the test set should be larger than one to guarantee that the self-supervised contrastive learning can be conducted for test-time adaptation. However, it can still be easily extended to the online manner where the test samples arrive one by one. First, a set of training samples are selected from the training set, replacing the $A(x)$ in Eq. 7. Then, after performing the optimization for each test sample, we replace one training sample in $A(x)$ with this test sample for future test samples. This transfers the whole pipeline into an online manner and allows the model to gradually adapt to the distribution of test data. We will add a more detailed discussion on this part in the revised version.
>
> **How to choose feature space.** Thanks for the valuable question. Given a specific model, we can decouple the model into two parts. One is the feature encoder that maps the raw input image into a latent embedding, while the other is the classifier that further maps the latent embedding into a probability vector among classes. Usually, for vision tasks, the classifier is composed of several fully-connected layers and a softmax layer as its final layer. We can select the features fed into the classifier, which is commonly in existing works [1,2], for contrastive learning. We will add a detailed specification on this in the revised version.
>
> [1] Supervised contrastive learning. In NeruIPS 2020.
>
> [2] Model-Contrastive Federated Learning. In CVPR 2021.

---

> > ### Comment · Reviewer_CniS · 2023-08-17
> >
> > Thank you for the detailed responses. After explanation, the difference between test-time shift and intra-client heterogeneity, and the difference between label-level and feature-level test-time shifts become very clear. I don't have any concerns, will keep my original score of 8 and recommend its acceptance.

---

### Official Review · Reviewer_9n3S · 2023-07-06

**Soundness:** 4 excellent
**Presentation:** 3 good
**Contribution:** 4 excellent
**Rating:** 8
**Confidence:** 5

**Summary:**

This paper focuses on the federated learning (FL) scenario with the test-time shift problem, which is a practical yet challenging research topic. With empirical study, the authors find that the inter-client heterogeneity in personalized FL can be further leveraged to build a robust FL framework against the test-time shift problem. Motivated by this, the authors propose a novel FL approach termed FedICON that uses train-time and test-time contrastive learning to boost the ability of local models to learn invariant information among inter-client heterogeneity caused by test-time shifts. Extensive experiments show the effectiveness of FedICON on FL scenarios with various test-time shifts.

**Strengths:**

Strengths
1. Valuable research problem. This paper addresses a new research problem, namely federated learning with feature-level test-time shifts. The exploration of this research topic has significant potential for application in various industry scenarios. The authors discuss the problem from the perspective of inter- and intra-client heterogeneity, providing valuable insights for future works.


2. Novel method. The proposed method, FedICON, is innovative in addressing the FL problem with test-time shifts. The incorporation of supervised and unsupervised contrastive learning during the training and testing phases to acquire invariant knowledge is a meaningful approach.


3. Extensive experiments. The authors have conducted an extensive set of experiments to evaluate the effectiveness of the proposed method. They have considered multiple scenarios, such as covariate shift, domain shift, attribute shift, etc, which enhance the persuasiveness of the experiments. The experimental results demonstrate a significant performance improvement achieved by FedICON.

**Weaknesses:**

Weaknesses:

1. Lack of discussion for the motivated experiment. In Sec 1, the authors present evidence that highlights the positive impact of inter-client heterogeneity in FL when addressing test-time shift problems. However, the paper lacks a thorough discussion of the underlying reasons behind this observation. It would be beneficial for the authors to provide deeper insights and further explanations regarding the reasons or deductions supporting this finding.

2. Design motivation needs to be more clear. Contrastive learning plays a crucial role in the proposed method. However, the authors have not sufficiently discussed why they choose contrastive learning to provide the pivot supervision signals instead of other supervised or unsupervised objectives, such as cross-entropy or self-reconstruction. It is important for the authors to provide a clearer explanation and justification for this choice, allowing readers to understand the design motivation of using contrastive learning in the proposed method.

3. There are some instances where unexpected periods (".") appear at the end of subheadings in certain subsections, such as Sec 3.1, 3.2, and 3.3. The authors should carefully review the paper to identify and correct any typos or format errors throughout the manuscript.

**Questions:**

As I pointed out in "Weaknesses", more discussions are expected:
1. The discussion for the motivated experiment.
2. The motivation for using contrastive learning in FedICON.


**Limitations:**

The authors have discussed the limitations of this work.

---

> ### Author Rebuttal · Authors · 2023-08-10
>
> **Lack of discussion for the motivated experiment.** Thanks for the valuable comments. In the paper, we empirically prove and illustrate the fact that inter-client heterogeneity in FL can help alleviate the test-time shift problem. As for the underlying reason for that, the naturally-existing heterogeneity among FL clients potentially improves the generalization ability of the model when data of diverse feature distributions contribute to the learned model. Moreover, the inherent heterogeneity allows the model to capture invariance among heterogeneous feature distributions by exploring the common feature/representation properties. We will add more discussion and explanation regarding the reasons and insights in the revised version.
>
> **More explanation about the motivation.** Thanks for the valuable comments. We utilize contrastive learning to provide pivot supervision signals because contrastive learning is designed to learn representations that are invariant to different transformations of the same instance which is aligned with our target to extract local invariance. As for the other types of objectives or learning pipelines, some of them lack discriminative ability, e.g., self-reconstruction, which makes them less useful for distinguishing between different instances. Also, some of them lack the ability to extend to unsupervised learning scenarios, e.g., cross-entropy, making them less flexible during the inference stage. We will add more explanation about our motivation to use contrastive learning in the revised version.
>
> **Some typos.** Thanks for the valuable comments. We will correct these typos in the revised version.

---

> > ### Comment · Reviewer_9n3S · 2023-08-18
> > **thanks for the response**
> >
> > Thanks for the response. All my concerns have been properly addressed. I will raise my score.

---

### Author Rebuttal · Authors · 2023-08-10

We thank the reviewers for their valuable comments. We are glad that the reviewers found that the problem we are solving is valuable and practical in federated learning (Reviewers 9n3S, CniS, v7Ae); our idea of leveraging inter-client heterogeneity to handle test-time shift problem is novel and insightful (Reviewer 9n3S, CniS, pJGu); our experiments are comprehensive and extensive (Reviewer 9n3S, CniS, pJGu). Both Reviewer CniS and Reviewer v7Ae think the paper is well-written and easy to understand. We respond to some shared concerns by more than one reviewer below. Detailed responses to each reviewer are provided in the following. We will incorporate all the feedback in the final version.

**C1. The case where test samples arrive one-by-one (Reviewer CniS and Review v7Ae).** Thanks for the your valuable comments and questions. In our current method, according to Eq. 7, the size of the test set should be larger than one to guarantee that the self-supervised contrastive learning can be conducted for test-time adaptation. *It does not require access to the full test dataset to conduct the test-time adaptation.* Moreover, it can still be easily extended to the online manner where the *test samples arrive one by one*. First, a set of training samples are selected from the training set, replacing the $A(\mathbf{x})$ in Eq. 7. Then, after performing the optimization for each test sample, we replace one training sample in $A(\mathbf{x})$ with this test sample for future test samples. This transfers the whole pipeline into an online manner and allows the model to gradually adapt to the distribution of test data. We will add a more detailed discussion on this part in the final version to clarify the property of testing scenarios.

---

### Author Response · Authors · 2023-08-14

Dear Reviewers and AC,


Sincerely thank you for reading our rebuttal.

We have tried our best to address most if not all concerns raised by the reviewers. Please let us know if you have any further questions about our paper or the rebuttal. Thank you.

Best regards,

Authors

---

### Decision · Program_Chairs · 2023-09-21

**Decision:**

Accept (poster)

**Comment:**

In the submitted manuscript, the authors present a contrastive learning-based Federated Learning (FL) framework designed to address test-time shifts. Experimental results on multiple datasets substantiate the method's effectiveness.

Post-rebuttal reviews reveal mixed opinions. Concerns have been raised about the paper's handling of inter-client heterogeneity and its comparative analysis with methods that combine both test-time shift and domain generalization techniques. My own careful review indicates that the authors satisfactorily address the inter-client heterogeneity issue. Regarding the lack of certain comparative baselines, I find the authors' focus on the novel setting of test-time shift in FL frameworks to be justifiable, and consider the exploration of combining domain generalization with test-time shifts as a topic for future work.

Given these considerations, I recommend acceptance of the paper.